# ORCAE-AOCC: A Centralized Portal for the Annotation of African Orphan Crop Genomes

**DOI:** 10.3390/genes10120950

**Published:** 2019-11-20

**Authors:** Anna E. J. Yssel, Shu-Min Kao, Yves Van de Peer, Lieven Sterck

**Affiliations:** 1Centre for Microbial Ecology and Genomics, Department of Biochemistry, Genetics and Microbiology, University of Pretoria, Pretoria 0028, South Africa; anna.yssel@up.ac.za; 2Centre for Bioinformatics and Computational Biology, Department of Biochemistry, Genetics and Microbiology, University of Pretoria, Pretoria 0028, South Africa; 3VIB-UGent Center for Plant Systems Biology, Technologiepark, Zwijnaarde 71, 9052 Ghent, Belgium; shu-min.kao@psb.vib-ugent.be (S.-M.K.); lieven.sterck@psb.vib-ugent.be (L.S.); 4Department of Plant Biotechnology and Bioinformatics, Ghent University, 9052 Ghent, Belgium

**Keywords:** genome portal, genome resource, genome annotation, manual curation, African orphan crops

## Abstract

ORCAE (Online Resource for Community Annotation of Eukaryotes) is a public genome annotation curation resource. ORCAE-AOCC is a branch that is dedicated to the genomes published as part of the African Orphan Crops Consortium (AOCC). The motivation behind the development of the ORCAE platform was to create a knowledge-based website where the research-community can make contributions to improve genome annotations. All changes to any given gene-model or gene description are stored, and the entire annotation history can be retrieved. Genomes can either be set to “public” or “restricted” mode; anonymous users can browse public genomes but cannot make any changes. Aside from providing a user- friendly interface to view genome annotations, the platform also includes tools and information (such as gene expression evidence) that enables authorized users to edit and validate genome annotations. The ORCAE-AOCC platform will enable various stakeholders from around the world to coordinate their efforts to annotate and study underutilized crops.

## 1. Introduction

According to the United Nations (2019), the number of undernourished people has increased in the past three years, with more than 820 million people still facing starvation. Moreover, the majority of these people live in developing countries. As the global population continues to grow, it is predicted that there will be 9 billion people by 2050, and the demand for food will be 70% greater than it is today [1]. Currently, out of the estimated 50,000 edible plant species, just three of them (maize, rice, and wheat) provide two-thirds of the world’s food energy intake. Technological innovations such as the development and selection of high-yield varieties, improvement of pesticide and fertilizer use, mechanization, and irrigation facilities have contributed to a global increase in the production of these grains. However, these innovations are often not readily available to the subsistence farmers throughout Africa. 

Apart from maize, rice, and wheat, which are all important for Africa and African farmers, there are many other crops that are currently underutilized but have great potential for Africa. These underutilized or so-called “orphan-crops”, are ancient, neglected, or indigenous crops with limited cultivation at a global scale, and their use ranges from food, fodder, to derivatives such as oil and medicine. In addition, orphan crops are promising solutions for nutritional diversity and can reduce over-reliance on major-crops and certain agricultural practices that have a negative environmental impact, as discussed by Mayes et al. [2]. In order to fast-track the improvement of orphan-crops, either by selective breeding or genetic modification, it is essential to have reliable information about their genetic makeup.

The African Orphan Crop Consortium (AOCC) was established in 2011 to tackle hunger and malnutrition in Africa [3]. AOCC aims to facilitate the development of locally available crops to supply nutritious and high yielding varieties. AOCC has committed itself to sequence, annotate, and analyze the genomes of 101 mostly indigenous and some introduced crops [3]. Given that orphan crops are expected to advance healthy food systems, as well as genetic resources for future crops, and agricultural sustainability under climate change [4], it is anticipated that the comprehensively selected 101 species in AOCC will become an invaluable resource and broaden the diversity of our current understanding of crop genomics (Figure 1).

The availability of high-quality crop reference genomes has already proved to be valuable to breeders, for example, by facilitating the identification of breeding targets in the genome [5]. Nevertheless, the complexity of many plant genomes, due to size, high repeat content, and polyploid ancestry has rendered their de novo genome assemblies particularly difficult [6]. Furthermore, different sequencing platforms create different challenges in the downstream bioinformatics analyses of a Next Generation Sequencing (NGS) project workflow [7,8,9]. On the other hand, long-read sequencing techniques and the recent advances of assemblers have generally increased the quality of draft genomes. However, fewer developments were made in terms of the genome annotation procedures, and some even argue that the errors of genome annotations keep propagating [9]. Here, we present a genomic resource called ORCAE-AOCC, a community-based genome annotation platform, and discuss its potential value for the scientific community.

Genome annotation mainly consists of two phases. (1) Structural annotation aims to provide information on the location of genes in the genome and the exact boundaries of exons and introns. The process often involves making use of transcript evidence [10,11] and ab initio modeling that utilizes statistical models to predict gene structures [11,12]. (2) Functional annotation aims to assign biological functions to the genes. Functional annotation is often homology-based, which means that information about the biological role of a gene in a new genome is inferred from genes with similar sequences from other genomes, where the role of the gene has been described or predicted [11]. Hypothetical genes (experimentally uncharacterized genes) or predicted gene models without any similarity found can sometimes be species-specific and their functions can be hard to deduce [13,14,15,16]. However, as more genomes are sequenced and annotated, it is becoming clear that some hypothetical genes are conserved between species [13]. Experimental evidence has shown that many hypothetical genes are indeed expressed and that they have critical biological roles [13,17,18,19]. Therefore, there is a need to focus efforts toward understanding the roles of hypothetically functional genes. Furthermore, it is also important to efficiently identify and exclude miss-annotated hypothetical genes [16].

Manual curation of genome annotations has been proven to be extremely valuable for building accurate reference gene sets in model organisms but compared to automatic annotation methods it is prohibitively expensive and is thus not widely done for non-model organisms [20]. Community- based efforts, where different researchers working on the same genome can contribute new information on gene structure and function have proven to be highly beneficial, e.g. in the case of the Vertebrate Genome Annotation Database (VEGA, [21]), which is maintained by the human and vertebrate analysis and annotation (Havana) team at the Wellcome Trust Sanger Institute (WTSI, [22]). Some of the best known community annotation platforms for plants include: The Arabidopsis Information Resource (TAIR) [23,24], the Maize Genetics and Genomics Database (Maize GDB) [25,26], The Rice Annotation Project Database (RAP-DB) [27,28,29], the International Wheat Genome Sequencing Consortium (IWGSC) [30,31] and Wheat@URGI [32,33]. Each of these platforms are dedicated to a single species that is widely grown and well-studied. Typically, these genomes have been annotated using automatic methods followed by manual curation steps [24,28]. Members of the scientific community can provide the curators of the databases above with information on newly identified genes and gene functions, which are then added when the genome annotations are updated.

## 2. ORCAE-AOCC and the Currently Deployed Genomes

Referred to as the Online Resource for Community Annotation of Eukaryotes, ORCAE [34] was developed and launched in 2012 to facilitate the manual curation of gene models, functional annotations, and improvement of annotation quality by genome consortia [35]. ORCAE was chosen as a central platform for the annotation of the grapevine (*Vitis vinifera*) as part of the International Grapevine Genome Project [36], In recent years, ORCAE has also been used by other communities resulting in several high-profile publications including the genomes of the seagrass (*Zostera marina*) [37], olive tree (*Olea europaea var. sylvestris*) [38], and sea lettuce (*Ulva mutabilis*) [39]. The system was designed with a wiki-like style editing mode for the community to refine information about the gene models, such as the gene structures and the definition of gene function. ORCAE also seamlessly integrates with GenomeView [40] to allow manual editing of the structure of a given gene model, with the aid of RNA-Seq or (expressed sequence tag) EST evidence. After editing the gene structures, the system will perform a number of checks in order to validate the modified gene structure before it is committed back to the database.

On the gene page (Figure 2), ORCAE displays the alignment of homologs retrieved from other public databases for a given gene, as well as other evidential information such as EST alignments and expression profiles. The built-in backend utility will search the detail properties of the protein sequence and provide protein domain information using InterProScan [41]. As an information collector, ORCAE allows expert annotators from the consortium to assess and update the information of each gene. Additionally, the functional description of each gene is transferred from the trusted reference database by homology and summarized using tools such as Automated Assignment of Human Readable Descriptions (AHRD) [42]. The system will automatically update the information of the gene after any modification. The final quality of the annotation depends not only on the initial deployed ab initio prediction but also on the effort of the consortium. Although it is hard to be scaled in all genome projects, it still provides a way to improve the quality of gene prediction and avoid the propagation of false positively predicted structures.

In order to incorporate the incoming genomes from the AOCC project into the ORCAE platform, while maintaning focus on the project, we launched ORCAE-AOCC [43], a dedicated genome portal for the AOCC consortium (Figure 3). ORCAE-AOCC currently contains five published genomes from the consortium: *Faidherbia albida*, *Moringa oleifera*, *Sclerocarya birrea*, *Lablab purpureus*, and *Vigna radiata* [44] the remaining genomes will be added as soon as their sequencing and annotation are complete. Apart from the curation of functional and structural annotation, users can also visit the portal of the desired genome in order to download the latest version of the annotations, the coding sequences (CDS) and the predicted protein sequences. Users can also perform BLAST [45] searches against CDS, protein, or the genomic sequence from within the portal. For on-going and restricted genome projects, one can request an account from the coordinator of the genome project to join the curation process prior to publication.

## 3. Concluding Remarks and Future Perspectives

ORCAE-AOCC is a platform that acts as a portal where the genomes that are sequenced as part of the AOCC initiative can be accessed by the broader research community. It also facilities collaborative efforts to improve the annotations and serves as a central repository for up to date versions of the genomes. ORCAE-AOCC also contains functionalities such as BLAST (Basic Local Alignment Search Tool) [46], the visualization of gene expression profiles, and pre-computed functional information. Additional African orphan crop genomes will be added when they will become available.

## Figures and Tables

**Figure 1 genes-10-00950-f001:**
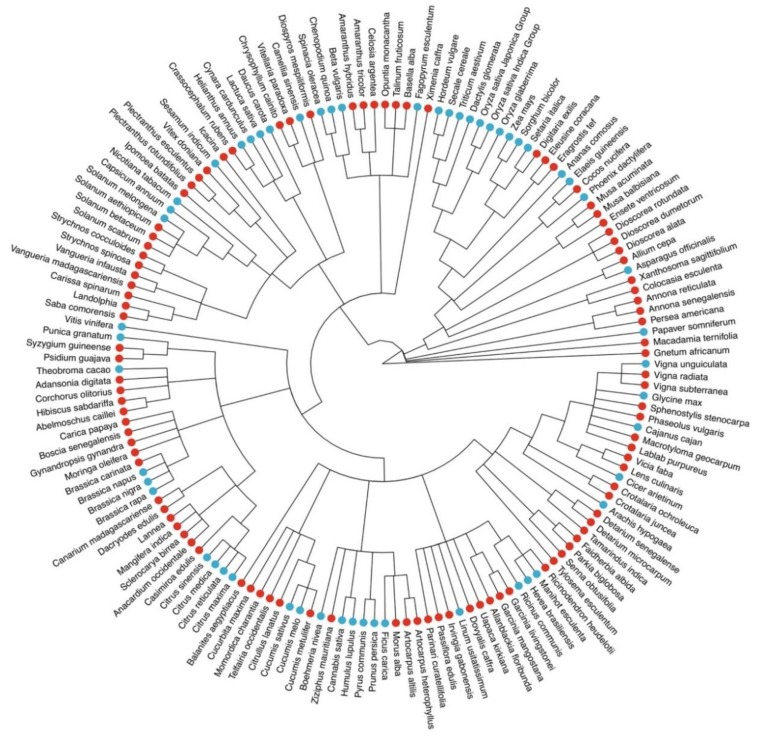
Diversity of currently sequenced, publicly available crop genomes (blue dots) and the orphan crop species initially included in the African Orphan Crops Consortium list (red dots), some of which are available on ORCAE already.

**Figure 2 genes-10-00950-f002:**
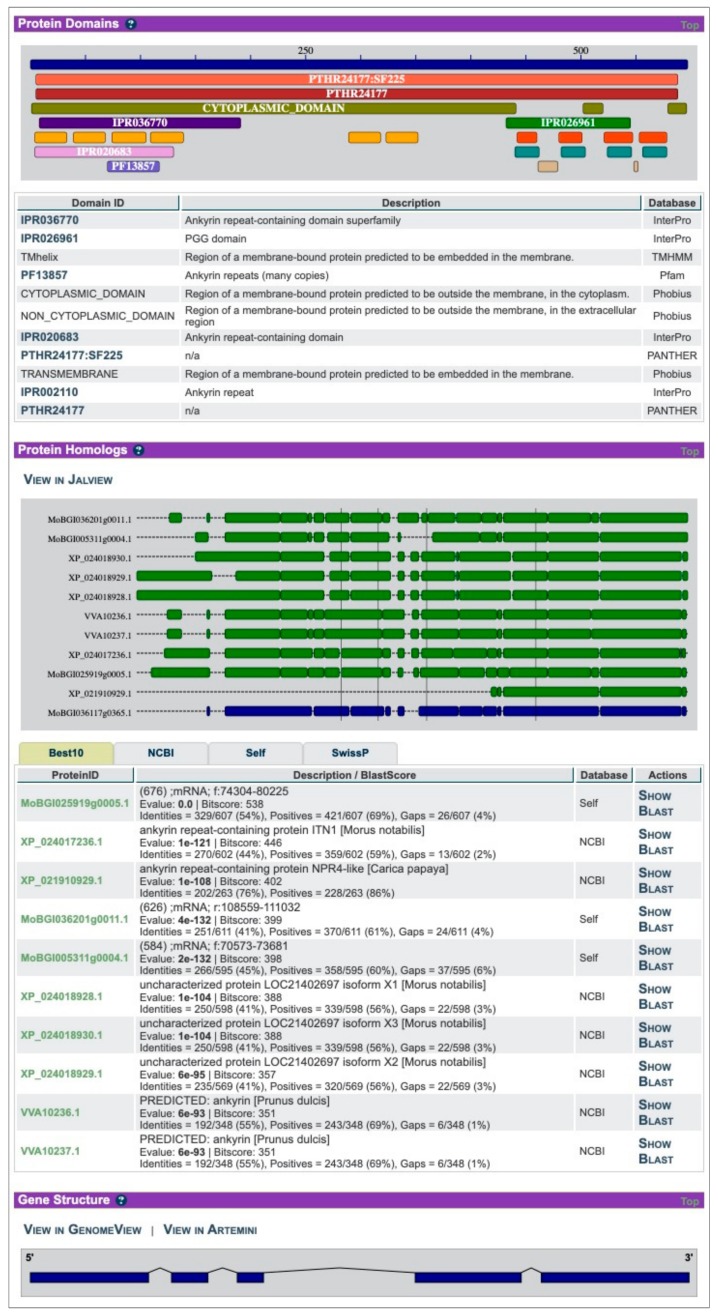
Part of the gene page in ORCAE-AOCC. Screenshot taken from [43].

**Figure 3 genes-10-00950-f003:**
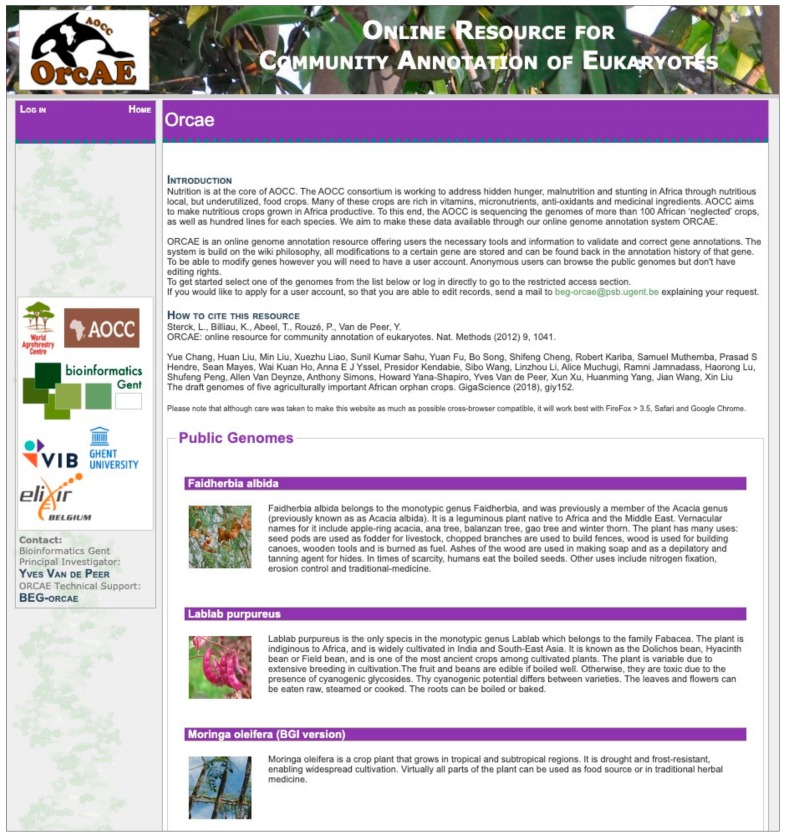
An overview of the ORCAE-AOCC genome portal.

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
