# Peer review of "ORCAE-AOCC: A Centralized Portal for the Annotation of African Orphan Crop Genomes"

_genes, 2019, doi:10.3390/genes10120950_

Round 1

Reviewer 1 Report

Yssel et al. introduce the features of ORCAE-AOCC as a centralized portal for the annotation of African Orphan Crop genomes. The manuscript is well written and clear. I appreciate the authors' efforts on building this annotation platform, which will benefit the community.

Author Response

The reviewer did not provide any points that need to be addressed. We are very pleased to read that Reviewer 1 appreciates our effort to develop the ORCAE platform.

Other changes:
Author affiliations have been updated (Anna Yssel omitted one of her affiliations in the previous version)
Yves Van De Peer should also be a corresponding author (along with Lieven Sterck)

Please note:
Anna Yssel and Shu-Min Kao are sharing the first author position as both have contributed equally to the work.

Reviewer 2 Report

This is a very well written paper, with clear descriptions of the tool and the purpose, and an incredibly helpful tool for the community! Thank you! 

Minor revisions as follows:

In the introduction you mention that 101 genomes will be sequenced. Then in figure 1 you show that around 50 have been sequenced so far, indicated by the blue dots. However the platform currently only has six genomes available, with two restricted. It would be nice mentioning somewhere the discrepancy between these numbers. Will all 101 genomes be sequenced and uploaded eventually? And if not, why only these six? Is there a rationale behind this? 

Very minor changes:

Line 29 - I would change "being hungry" to "facing starvation".  Line 112 - Typo. Should be "the system will perform". Line 132 - Change "Apart from" to "As well as"

Author Response

Point 1:
In the introduction, you mention that 101 genomes will be sequenced. Then in figure 1, you show that around 50 have been sequenced so far, indicated by the blue dots. However the platform currently only has six genomes available, with two restricted. It would be nice mentioning somewhere the discrepancy between these numbers. Will all 101 genomes be sequenced and uploaded eventually? And if not, why only these six? Is there a rationale behind this? 

Response 1:
The reviewer observed that out of the 101 crops that are on the AOCC list, only 6 (in fact, 5) are available on the ORCAE platform. The reason for this is that not all the species have been sequenced, assembled, and annotated yet. As soon as they are complete they will be added to the ORCAE platform. A sentence was added to the main text to indicate this, and the caption was also modified accordingly.

Point 2:
Line 29 - I would change "being hungry" to "facing starvation".  Line 112 - Typo. Should be "the system will perform". Line 132 - Change "Apart from" to "As well as"

Response 2:
The reviewer pointed out 3 lines containing grammatical errors (in original documents they were located on line 29, 112 and 132). We have changed line 29 and 112 according to the reviewer's suggestion. Line 132 was modified from "Apart from the curation of functional and structural annotation, users can also visit the portal of the desired genome to download the latest version of the annotations, as well as  CDS and protein sequences. " to "Apart from the curation of functional and structural annotation, users can also visit the portal of the desired genome order to download the latest version of the annotations, as well as the CDS and the predicted protein sequences."

Other changes:
Author affiliations have been updated (Anna Yssel omitted one of her affiliations in the previous version)
Yves Van De Peer should also be a corresponding author (along with Lieven Sterck)

Please note:
Anna Yssel and Shu-Min Kao are sharing the first author position as both have contributed equally to the work.